# Deciphering the late steps of rifamycin biosynthesis

Feifei Qi[1], Chao Lei[2], Fengwei Li[1], Xingwang Zhang[1], Jin Wang[2], Wei Zhang[1], Zhen Fan[2], Weichao Li[2], Gong-Li Tang [3], Youli Xiao [2,4], Guoping Zhao[2,4] & Shengying Li[1,4]

Rifamycin-derived drugs, including rifampin, rifabutin, rifapentine, and rifaximin, have long been used as first-line therapies for the treatment of tuberculosis and other deadly infections. However, the late steps leading to the biosynthesis of the industrially important rifamycin SV and B remain largely unknown. Here, we characterize a network of reactions underlying the biosynthesis of rifamycin SV, S, L, O, and B. The two-subunit transketolase Rif15 and the cytochrome P450 enzyme Rif16 are found to mediate, respectively, a unique C–O bond formation in rifamycin L and an atypical P450 ester-to-ether transformation from rifamycin L to B. Both reactions showcase interesting chemistries for these two widespread and well-studied enzyme families.

[1] Shandong Provincial Key Laboratory of Synthetic Biology, CAS Key Laboratory of Biofuels, Qingdao Institute of Bioenergy and Bioprocess Technology, Chinese Academy of Sciences, Qingdao, Shandong 266101, China. [2] CAS Key Laboratory of Synthetic Biology, Institute of Plant Physiology and Ecology, Shanghai Institutes for Biological Sciences, Chinese Academy of Sciences, 200032 Shanghai, China. [3] State Key Laboratory of Bio-Organic and Natural Products Chemistry, Shanghai Institute of Organic Chemistry, Chinese Academy of Sciences, 200032 Shanghai, China. [4] University of Chinese Academy of Sciences, 100049 Beijing, China. These authors contributed equally: Feifei Qi, Chao Lei. Correspondence and requests for materials should be addressed to Y.X. (email: ylxiao@sibs.ac.cn) or to S.L. (email: lishengying@qibebt.ac.cn)

Rifamycins are ansamycin antibiotics that show a wide spectrum of antimicrobial activities against both Gram-positive and Gram-negative bacteria[1]. Their semisynthetic derivatives such as rifampin, rifabutin, rifapentine, and rifaximin have been used for decades in the clinic for the treatment of tuberculosis, leprosy, and AIDS-related mycobacterial infections[2], and their recognized pharmacological mode-of-action is the specific inhibition of prokaryotic DNA-dependent RNA synthesis[2,3].

Floss and co-workers discovered the first rifamycin biosynthetic gene cluster comprised of 34 genes in the bacterium *Amycolatopsis mediterranei* S699[4]. Five type I polyketide synthases (PKSs: RifA-E) encoded by the cluster are responsible for assembling the first macrocyclic intermediate (proansamycin X) using 3-amino-5-hydroxybenzoic acid as the starter unit, and malonyl-CoA and methylmalonyl-CoA as extender units[5,6]. Further post-PKS modifications, including the dehydrogenation of C-8 and the hydroxylation of C-34, lead to the intermediate rifamycin W[7]. Subsequently, Rif5 converts the Δ12,29 olefinic bond of rifamycin W into a ketal moiety, followed by Rif20-mediated acetylation of the hydroxyl group at C-25 and Rif14-mediated *O*-methylation at C-27, producing rifamycin SV (R-SV) (Supplementary Fig. 1)[2,8–10]. The later biosynthetic steps leading from R-SV to the end product rifamycin B (R-B) (Fig. 1) have not been biochemically characterized.

During fermentation, R-B is the predominant rifamycin product accumulated by wild type *A. mediterranei* and by the first industrial strains[11]. However, as the antibacterial activity of R-B is modest, R-B needs to be transformed back to the more bioactive R-SV before being subjected to the chemical, enzymatic, or biotransformation process that yield multiple highly potent clinical drugs[2]. Consequently, strains that produce high levels of R-SV are now preferred by industry (e.g., the well-studied mutant strain *A. mediterranei* U32). Nonetheless, the strain improvement for R-SV high-producers has not been as successful as that for R-B producing strains[2], so both kinds of strains are still required by industry. Notably, rifamycin S (R-S), rifamycin L (R-L), and rifamycin O (R-O) are also important intermediates that can be prepared from *A. mediterranei* fermentation cultures[11–14]. However, the biosynthetic relationship between these late rifamycin derivatives remains elusive, despite previous research attempts based on extensive isotopic feeding and mutagenesis experiments[2,13,15].

Recently, comparative analysis of the rifamycin biosynthetic gene clusters of *A. mediterranei* S699 (an R-B producer) and U32 (an R-SV producer), together with corresponding genetic complementation testing, strongly suggested that the cytochrome P450 enzyme Rif16 is involved in the conversion of R-SV to R-B[16]. Furthermore, gene inactivation and complementation experiments showed that Rif16 and the two-subunit transketolase Rif15 (encoded by *rif15a* and *rif15b*, two overlapping genes, Supplementary Table 1) are essential and sufficient for this transformation[17]. However, the biochemical and chemical mechanisms underlying the unusual ether bond formation between the C-4 phenolic hydroxyl group of R-SV and a glycolic acid moiety leading to R-B remain unknown. Here, by reconstituting the in vitro activity of Rif15 and Rif16, we reveal a biosynthetic network for the inter-conversion of R-SV, R-S, R-L, R-O, and R-B (Fig. 1), finally elucidating the mechanisms for the late reactions of rifamycin biosynthesis.

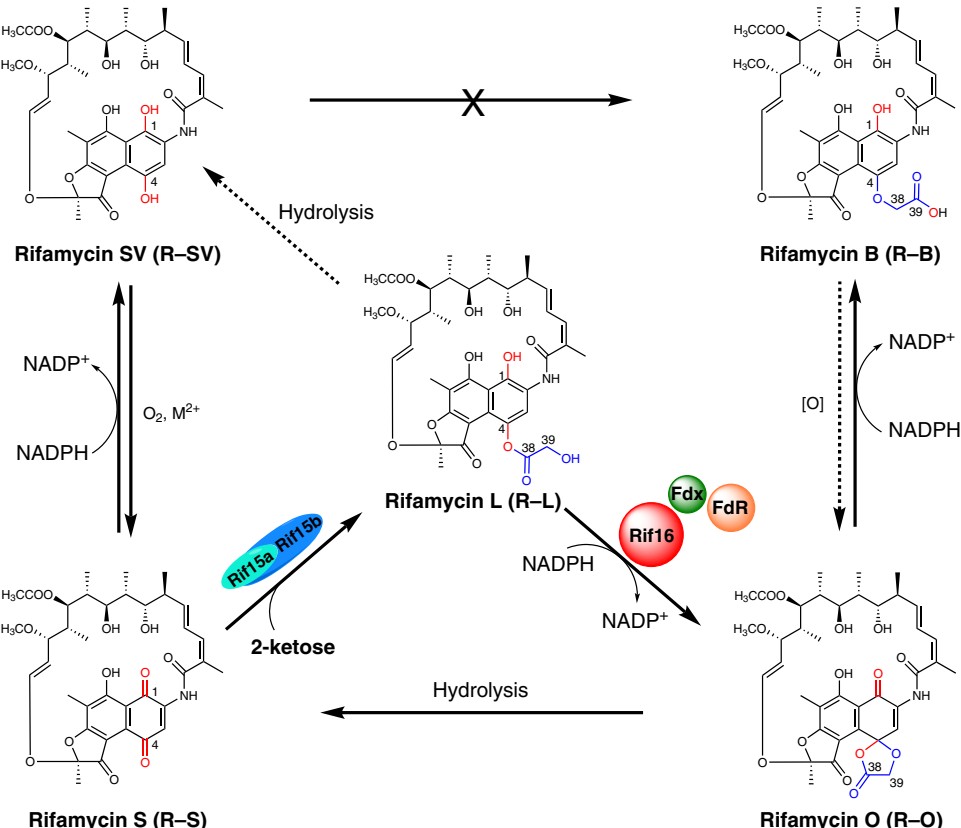

**Fig. 1** The biosynthetic network of late rifamycin derivatives. R-SV can be oxidized to R-S spontaneously in the presence of dioxygen and divalent metal ions. The transketolase Rif15 is responsible for transferring a $C_2$ keto-containing fragment from a 2-ketose to R-S, giving rise to R-L. The P450 enzyme Rif16 catalyzes the transformation from R-L to R-O in the presence of NADPH, ferredoxin (Fdx), and ferredoxin reductase (FdR). Finally, R-O is non-enzymatically reduced to R-B by NADPH

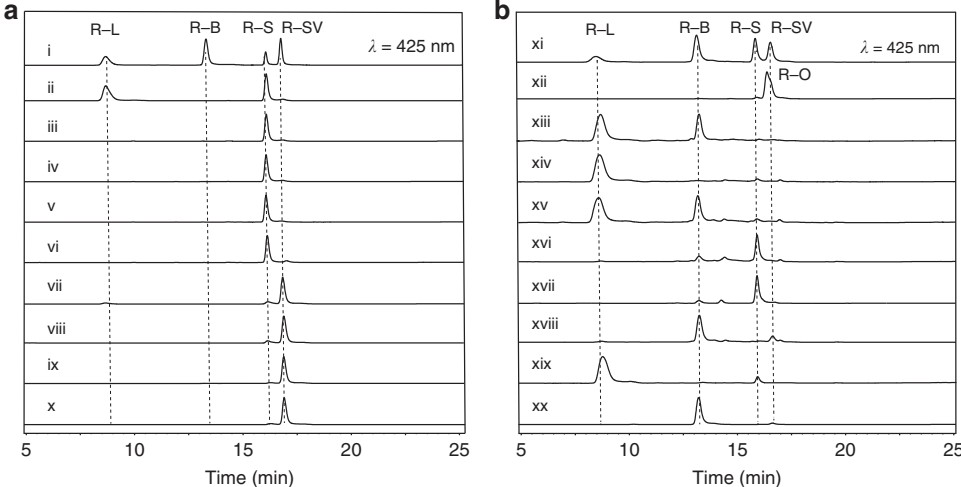

**Fig. 2** HPLC analysis of the reactions catalyzed by Rif15 and Rif16. **a** Reactions catalyzed by the transketolase Rif15. (i) The mixed R-L, R-B, R-S, and R-SV standards; (ii) R-S with Rif15 in the presence of F-6-P, ThDP and $Mg^{2+}$; (iii–vi) the negative controls of (ii) with the omission of Rif15a (iii), Rif15b (iv), ThDP (v), or $Mg^{2+}$ (vi); (vii) R-SV with Rif15 in the presence of F-6-P, ThDP and $Mg^{2+}$; (viii) R-SV in reaction buffer; (ix) R-SV in reaction buffer with the addition of 2 mM ascorbic acid; and (x) R-SV with Rif15 in the presence of F-6-P, ThDP, $Mg^{2+}$, and 2 mM ascorbic acid. **b** Reactions catalyzed by the cytochrome P450 enzyme Rif16. (xi) The mixed R-L, R-B, R-S, and R-SV standards; (xii) The freshly prepared R-O authentic standard; (xiii) R-L with Rif16 in the presence of *se*Fdx, *se*FdR, and NADPH; (xiv) the negative control of (xiii) with the omission of NADPH; (xv) co-injection of (xiii) with 50 μM R-B; (xvi) R-L with Rif16 in the presence of 20 mM $H_2O_2$; (xvii) R-O in reaction buffer; (xviii) R-L with Rif16 in the presence of 20 mM $H_2O_2$ and 1 mM NADPH; (xix) R-L with 20 mM $H_2O_2$; (xx) R-O and 1 mM NADPH in reaction buffer

## Results and Discussion

**Initial examination of the enzymatic activity of Rif16**. Generally, P450 enzymes catalyze oxidative reactions[18,19] and transketolases can, in the presence of the essential cofactor thiamine diphosphate (ThDP), transfer a $C_2$ keto-containing fragment from a 2-ketose (e.g., fructose-6-phosphate (F-6-P), xylulose-5-phosphate (Xu-5-P), ribulose-5-phosphate (Ru-5-P), sedoheptulose-7-phosphate (S-7-P), dihydroxyacetone (DHA), etc.) to the first carbon atom of an aldose (e.g., ribose 5-phosphate, glyceraldehyde 3-phosphate, etc.)[20]. Based on the previous proposal that R-SV could be a biosynthetic precursor of R-B[2], we first surmised that Rif16 (CYP105G1[21]) may oxidize R-SV to R-S and that R-S (containing a C-4 keto group) may be a substrate of Rif15. To test these enzymatic hypotheses, we sub-cloned *rif15a*, *rif15b*, and *rif16* (Supplementary Fig. 2) and heterologously expressed these genes in *Escherichia coli* Codon Plus (DE3)-RIPL. We then used Ni-NTA chromatography to purify the *N*-terminally $His_6$-tagged Rif15a, Rif15b, and Rif16 to homogeneity (Supplementary Fig. 3). Notably, the *N*-$His_6$-tagged Rif15a and the non-tagged Rif15b were able to be co-expressed and co-purified, suggesting a strong interaction between these two Rif15 subunits (Supplementary Fig. 3).

Purified Rif16 appeared to be a functional P450 enzyme, as it had the expected red color and showed a signature peak at 450 nm in its CO-reduced difference spectrum (Supplementary Fig. 4). To reconstitute the in vitro activity of Rif16, we used two surrogate redox partner proteins to shuttle electrons from NADPH to the heme-iron reactive center for P450 catalysis: the ferredoxin *se*Fdx (SynPcc7942_1499) and the ferredoxin reductase *se*FdR (SynPcc7942_0978), both of which are from the cyanobacterium strain *Synechococcus elongatus* PCC 7942 and were here expressed heterologously in *E. coli* and purified[22]. Against our expectations, Rif16 was not able to catalyze the conversion from R-SV to R-S, while R-S was readily reduced to R-SV by addition of NADPH alone (Supplementary Fig. 5). Importantly, the hydroquinone R-SV was spontaneously oxidized to the quinone R-S by ambient $O_2$, and this transformation was dramatically accelerated by the presence of divalent metal ions

(e.g., $Cu^{2+}$, $Mn^{2+}$, etc.) (Supplementary Fig. 6), similar to previously reported findings[23]. However, we cannot exclude the possibility that an oxidase might be responsible for enzymatic oxidation of R-SV into R-S in vivo. Taken together, our results suggest that Rif16, rather than performing a normal bio-oxidation, may catalyze an atypical P450 reaction in rifamycin biosynthesis.

**Functional characterization of Rif15**. We next evaluated the in vitro activity of Rif15a/Rif15b at a 1:1 ratio (i.e., the reconstituted Rif15 transketolase) in the presence of R-S and F-6-P as the potential $C_2$ keto acceptor and donor, respectively, with ThDP and $MgCl_2$ as cofactors. As predicted, Rif15 converted R-S into a different product with higher polarity than R-B, while single subunits (that is, either Rif15a or Rif15b alone) were not able to catalyze the same transformation. Additionally, we found that both ThDP and $Mg^{2+}$ were required for the catalytic activity of Rif15 (Fig. 2a, trace i–vi). This is unsurprising since the diphosphate moiety of ThDP is bound to the transketolase through a bivalent cation to form the catalytically active *holo*-enzyme from the *apo*-protein[24,25]. Protein sequence alignment of multiple transketolases shows that the residues involved in the interactions with ThDP and the metal ion are highly conserved regardless their origins and subunit organization modes (Supplementary Fig. 7).

High performance liquid chromatography-high resolution mass spectrometry (HPLC-HRMS) analysis revealed that the $m/z$ value of the product was 754.3069 ([M-H]⁻, deduced to be $[C_{39}H_{48}NO_{14}]^-$) (Supplementary Fig. 8), which is consistent with that of R-L or R-B in negative ion mode (*calc.* 754.3069). Since the retention time of this product was distinct from that of R-B, we suspected that the product here is R-L. Both the 1D (¹H, ¹³C, DEPT135) and 2D (¹H-¹H COSY, HSQC, HMBC) NMR spectra (Supplementary Figs. 9–13, Supplementary Table 2) of the purified product were acquired, and spectral comparisons of the proton NMR data obtained from the product and the substrate R-S (Supplementary Fig. 9) revealed a new set of $CH_2$ signals ($\delta_{H-39}$

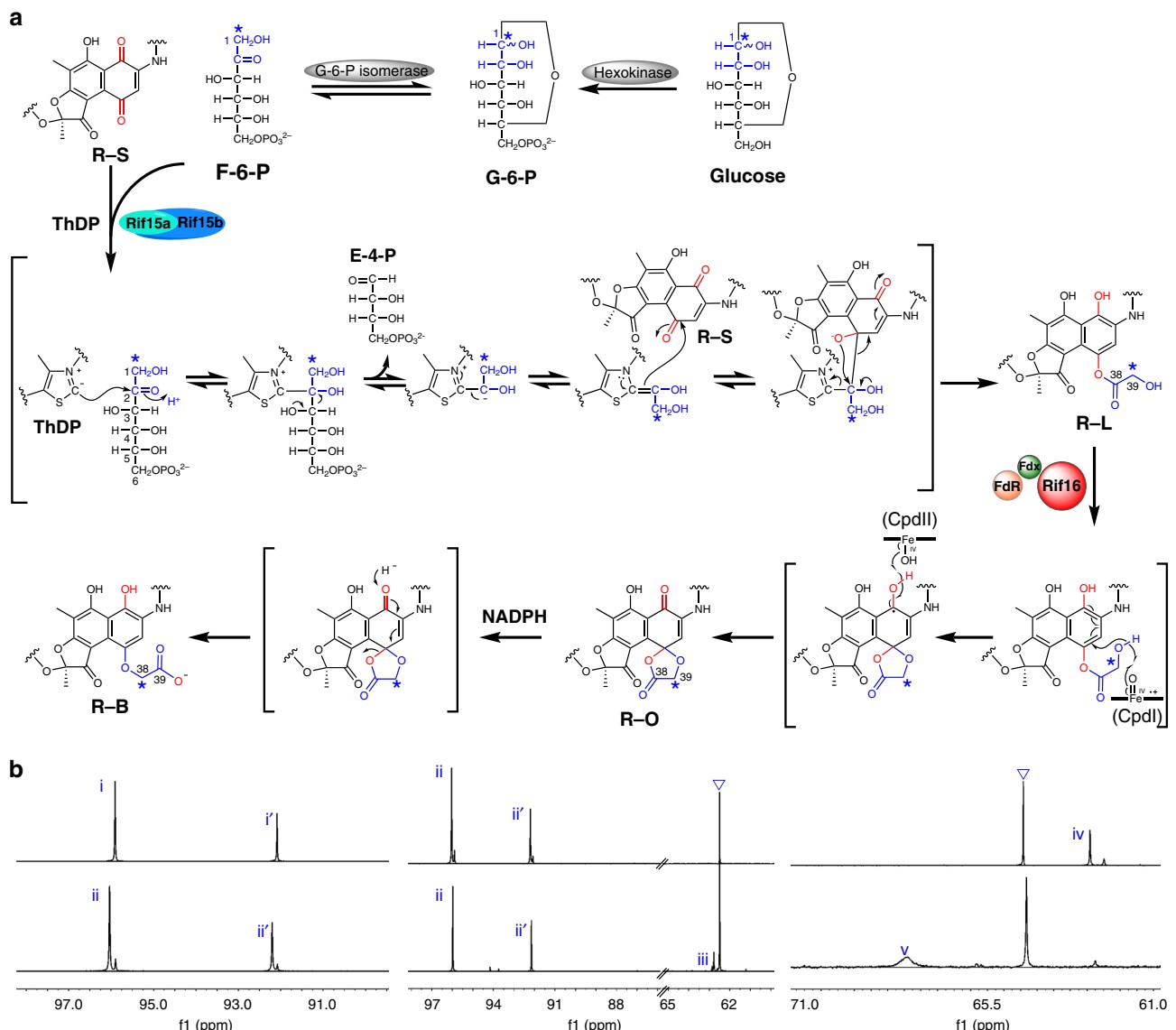

**Fig. 3** Putative mechanisms for Rif15 and Rif16 supported by $^{13}$C-tracer NMR results. **a** The proposed catalytic mechanisms for Rif15 and Rif16. The $^{13}$C labeled carbons marked by asterisks originate from [1-$^{13}$C]glucose. **b** The $^{13}$C NMR spectra of [1-$^{13}$C]($\pm$)-glucose (i, i'), [1-$^{13}$C] ($\pm$)-G-6-P (ii, ii'), [1-$^{13}$C]F-6-P (iii) in D$_2$O, and (39-$^{13}$C]R-L (iv) and [38-$^{13}$C]R-B (v) in CD$_3$OD. The triangles indicate the carbon signals of residual glycerol derived from enzymatic reaction buffer

4.63, $d$, $J = 17.1$ Hz, 1 H, $\delta_{H-39}$ 4.56, $d$, $J = 17.1$ Hz, 1 H vs $\delta_{H-38}$ 4.72, s, 2 H in R-B; $\delta_{C-39}$ 62.3 vs $\delta_{C-38}$ 67.8 in R-B) from the product. Further extensive analyses of 2D NMR correlations of this methylene group confirmed it to be R-L. The steady-state kinetic constants of Rif15 were determined with $K_m = 8.8 \pm 2.4$ μM and $k_{cat} = (2.2 \pm 0.1) \times 10^{-2}$ min$^{-1}$. Moreover, Xu-5-P, Ru-5-P, S-7-P, and DHA were also able to serve as alternative C$_2$ keto donors for Rif15, with Xu-5-P being optimal in terms of conversion ratios under the same conditions (Supplementary Fig. 14).

To examine whether R-SV is also a direct precursor of R-L as previously proposed[2], similar Rif15 reactions were performed using R-SV as substrate, and, interestingly, we detected a small amount of R-L as a product (Fig. 2a, trace vii). In light of the spontaneous oxidation from R-SV to R-S by O$_2$ that we had observed in earlier assays (in aqueous solution in the presence of Mg$^{2+}$, Supplementary Fig. 6), 2 mM ascorbic acid was added to the Rif15 reactions to protect R-SV from oxidation (Fig. 2a, trace viii, ix)[26]. Upon this addition, we no longer detected R-L as a

reaction product (Fig. 2a, trace x), establishing that the previously detected R-L was actually derived from the spontaneously formed R-S.

Collectively, our results from these in vitro assays demonstrate that Rif15 is a two-component transketolase that transforms a ketone (in R-S) into an ester (in R-L), a reaction that has not been found previously in natural product synthesis, to the best of our knowledge. Mechanistically, the deprotonation of ThDP at the thiazolium ring generates a carbanion, which is responsible for cleaving the C-2/C-3 bond in the 2-ketose. The resultant ThDP-bound dihydroxyethyl group then undertakes nucleophilic attack of the C-4 carbonyl carbon of R-S, which is followed by bond rearrangements and re-aromatization, ultimately yielding R-L (Fig. 3a).

**Biochemical, structural, and mechanistic characterization of Rif16.** Having characterized the R-SV→R-S→R-L transformation, we next sought to resolve the conversion of R-L into R-B. These

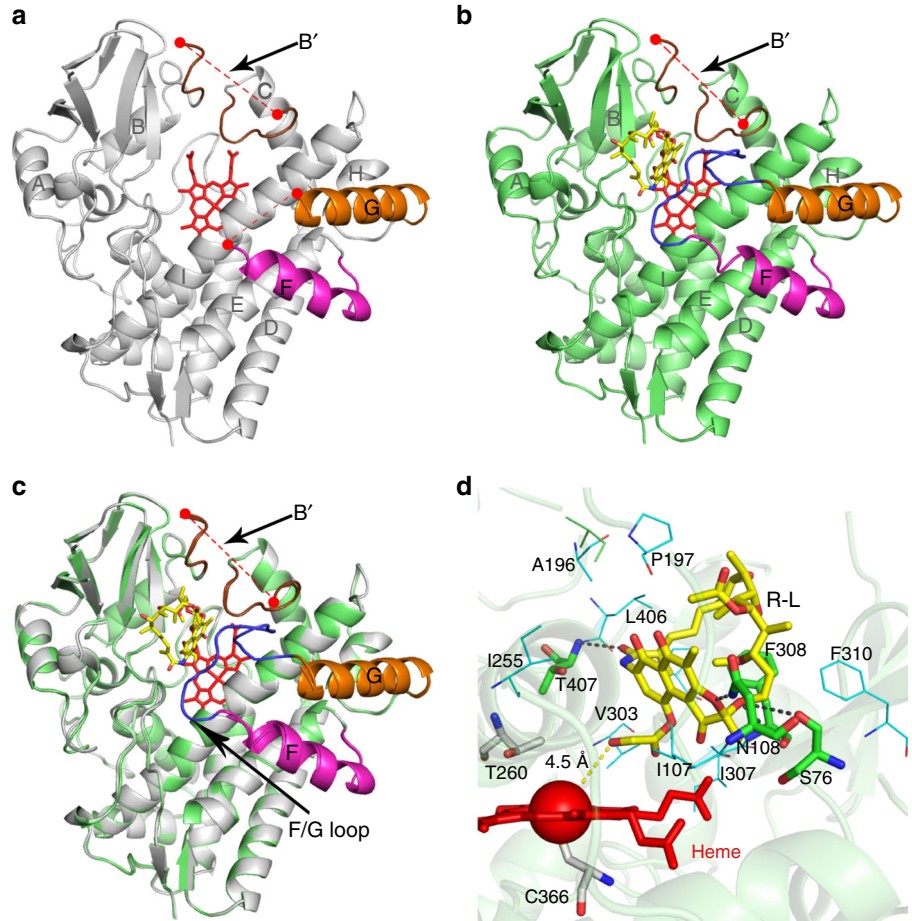

**Fig. 4** Structures of Rif16. **a** Substrate-free Rif16. The disordered B' loop (with 83–97 residues missing as shown with the dashed red line) that replaces the typical B' helix is shown in brown. The F and G helices are shown in magenta and orange, respectively. The disordered FG loop (192–205 residues) lacks electron density. The heme group is shown as a stick in red. **b** Substrate-bound Rif16. The B' loop in brown remains disordered. The FG loop (here ordered) is shown in blue. **c** Superimposition of the substrate-free (gray) and the substrate-bound (lime green) Rif16. The black arrows point out the regions that undergo conformational changes upon R-L binding. **d** The active site of Rif16. The substrate R-L and heme are shown as sticks in yellow and red, respectively, with the heme-iron depicted as a sphere. The key residues forming hydrogen bonds (black dashed lines) with R-L are shown as sticks in green. The residues within 5 Å around the substrate that constitute a large hydrophobic pocket are shown as lines in cyan. The conserved T260 and C366 are shown as sticks in silver. The distances (in angstroms) are indicated by the dashed yellow line

two rifamycin derivatives have the same oxidation state, but we still chose to test the activity of Rif16 against R-L, since this P450 enzyme was previously shown to be required for R-B biosynthesis[17]. Indeed, we found that R-L was significantly converted into R-B by Rif16 in the presence of seFdx, seFdR, and NADPH (Fig. 2b, trace xiii, xiv); the structure of the product was confirmed by the identical retention time of the product and the R-B authentic standard, co-elution with R-B in a co-injection experiment (Fig. 2b, trace xv), and the consistently observed $m/z$ value of 754.3069 ([M-H]$^-$, calc.754.3069) (Supplementary Fig. 15). These results clearly establish that Rif16 is the long-sought R-B synthase of rifamycin biosynthesis.

To elucidate the catalytic mechanism for this atypical ester-to-ether transformation, the crystal structures of substrate-free Rif16 (PDB ID code: 5YSM, Fig. 4a) and R-L-bound Rif16 (PDB ID code: 5YSW, Fig. 4b) were solved at 1.90 Å and 2.60 Å resolution, respectively. In both of the structures, there was only one typical cytochrome P450 fold existing in an asymmetric unit. The BB' loop-B' helix-B'C loop region, which is known to be important for substrate specificity determination[27], is significantly longer than those of many P450 enzymes that recognize smaller

substrates (Supplementary Fig. 16). The missing electron density of this region in both structures suggests the great structural flexibility. Both findings might help explain how Rif16 is able to accommodate its bulky substrate R-L, which represents one of the largest substrates for a P450 enzyme with the substrate-bound crystal structure available[28]. In the absence of substrate, Rif16 adopts an open conformation characterized by retraction of the F and G helices, loss of order in the B' helix, and missing electron density for the B'C and FG loops (Fig. 4a). A water molecule that is 2.5 Å away from the heme-iron forms the sixth axial ligand of Fe$^{3+}$ (Supplementary Fig. 17a). Upon binding with R-L, Rif16's FG loop becomes ordered but the B' helix and the B'C loop remain disordered (Fig. 4b), thereby adopting a partially open conformation rather than the closed conformation observed in many substrate-bound P450 enzymes[29,30] (Fig. 4c).

In the substrate-bound structure (Fig. 4d and Supplementary Fig. 17b), R-L forms hydrogen bonds with residues S76, N108, F308, and T407, and additionally interacts with residues I107, A196, P197, I255, V303, P305, I307, F310, and L406 via hydrophobic interactions (all of these residues are within 5 Å of R-L). Critically, the axial water ligand is displaced and the

hydroxyl group at C-39 of R-L is closest (4.5 Å) to the heme-iron reactive center. These structural features suggest a possible mechanism for R-B production (Fig. 3a): the ferryl-oxo species (Compound I) likely abstracts the hydrogen atom of the C-39 hydroxyl group, leading to formation of a substrate radical and the ferryl-hydroxo Compound II. The resultant oxygen radical can then directly attack the neighboring arene to form a five-membered ring pendant, and the radical would be delocalized to the aromatic ring. Next, the relocation of the spirocyclic intermediate could induce the second hydrogen abstraction from the C-1 hydroxyl group by Compound II. This diradical mechanism might result in the formation of R-O. Notably, similar mechanisms—involving two alternative substrate binding poses being responsible for hydrogen abstractions from two distant sites—have been proposed for C–O coupling reactions catalyzed by a number of P450 enzymes[31,32]. Finally, the pentabasic cyclic compound R-O could be reduced to R-B (rather than R-L) by the NADPH-derived hydride, since the carboxylic acid is a better leaving group than the alcohol.

This R-L→R-B conversion reaction lacks net oxidation-reduction. To dissect this unusual P450 reaction experimentally, we elected to oxidize R-L by taking advantage of the peroxide shunt pathway of Rif16[18], in which $H_2O_2$ acts as the sole oxygen and electron donor of Rif16; this approach allowed us to bypass the dual roles of NADPH from our previous reaction system (its roles as an electron donor for the P450 enzyme and as a hydride provider for direct reduction of R-O). Interestingly, R-S was the dominant product from this reaction (Fig. 2b, trace xvi), which likely resulted from the spontaneous hydrolysis of the P450 product R-O[15,33] (Fig. 2b, trace xvii). The addition of NADPH into the Rif16/R-L/$H_2O_2$ system led to predominant production of R-B (Fig. 2b, trace xviii, xix), as R-O can be reduced to R-B in the presence of NADPH (Fig. 2b, trace xx)[34,35]. Furthermore, the unstable compound R-O with the correct $m/z$ value of 752.2920 ([M−H]⁻, calc. 752.2924) was observed in a time-course study (Supplementary Fig. 18). These results strongly suggest that R-O is the intermediate that enables the conversion of R-L to R-B.

To validate our proposed enzymatic reaction mechanisms, we performed a series of $^{13}C$-tracer NMR experiments. First, [39-$^{13}C$]R-L was prepared by mixing [1-$^{13}C$]glucose, ATP, $Mg^{2+}$, hexokinase, G-6-P isomerase, Rif15a/Rif15b, ThDP, and R-S in a one-pot reaction. We observed that [1-$^{13}C$]glucose was phosphorylated to [1-$^{13}C$]G-6-P by hexokinase, which was subsequently transformed into [1-$^{13}C$]F-6-P by G-6-P isomerase (Fig. 3). The Rif15-mediated transfer of the $^{13}C$-labeled glycolic acid $C_2$ moiety from [1-$^{13}C$]F-6-P to R-S resulted in production of [39-$^{13}C$]R-L, with an enriched C-39 signal of $\delta_C$ 62.4 (Fig. 3b). The identity of this product was further confirmed by LC-HRMS analysis indicating an $m/z$ value of 755.3106 ([M−H]⁻, calc. 755.3105, Supplementary Fig. 19), which is ~1 Da greater than that of unlabeled R-L [M−H]⁻ = 754.3069). Next, Rif16, seFdx/ seFdR, and NADPH were added into the above one-pot reaction. As expected, the P450 enzyme converted [39-$^{13}C$]R-L into [38-$^{13}C$]R-B, confirmed by LC-HRMS with an $m/z$ value of 755.3100 ([M−H]⁻, calc. 755.3105, Supplementary Fig. 19) and by our observation that the $^{13}C$-labeled carbon signal shifted downfield from $\delta_C$ 62.4 to $\delta_C$ 67.9 (Fig. 3b); both analytical results are consistent with the conversion of R-L to R-B via R-O (Fig. 3a).

It was previously reported that the R-SV high-producer A. mediterranei U32 has an R84W single mutation in Rif16[16]. The understanding of Rif16 mechanism allowed us to rationalize this industrially important phenotype. Specifically, the dissociation constant ($K_d$) of R-L toward Rif16 was determined to be $1.3 \pm 0.1 \mu M$ (Supplementary Fig. 20), while the purified Rif16$_{R84W}$ mutant (Supplementary Figs. 3 and 21) showed no detectable binding of R-L and lost the ability of catalyzing the

transformation from R-L to R-B (Supplementary Fig. 22). Since R84 is located at the B' loop of Rif16 (Supplementary Fig. 16), which is an important region for P450 substrate recognition[27], its replacement by a tryptophan abolishes the productive substrate binding via a mechanism to be elucidated. Furthermore, according to the biosynthetic network shown in Fig. 1, the U32 mutant should accumulate R-L instead of the observed R-SV and R-S[16]. We reason that the ester R-L might be unstable, which could be hydrolyzed to R-SV either enzymatically or spontaneously (Supplementary Fig. 23).

Our elucidation of the network comprising the late steps of rifamycin biosynthesis revealed a unique C–O bond formation reaction mediated by a transketolase that involves both normal C–C bond formation and unusual bond rearrangements. Notably, transketolases primarily participate in central metabolic pathways such as pentose phosphate pathway and the Calvin cycle, and there have been few reports on transketolases that are involved in natural product biosynthesis[36,37]. The ether bond formation derived from the concomitant oxidation-reduction reactions and complex bond rearrangements also represents a highly atypical reaction for a P450 reaction system. The knowledge on the slow kinetics and the optimal $C_2$ keto donor of Rif15 could also help direct the future rational strain improvement. Finally, BLAST searches demonstrate that there exist other protein sequences with high similarity to Rif15 and Rif16 (Supplementary Table 3, Supplementary Fig. 24), suggesting that more Rif15-like and Rif16-like functionality could be further identified. Some of these enzymes come from rifamycin producing microorganisms[38–41], which may suggest an effective method for discovery of more rifamycin producers by using Rif15 and Rif16 sequences as probes.

## Methods

**Chemicals**. Rifamycin SV and rifamycin O authentic standards were purchased from Sigma Aldrich (USA) and Toronto Research Chemicals (Canada), respectively. Rifamycin S and rifamycin B authentic standards were bought from National Institutes for Food and Drug Control (China).

**General DNA manipulation**. The E. coli DH5α strain was used for plasmid construction, storage, and isolation. Fast-digest restriction endonucleases (Thermo Fisher Scientific, USA) and T4 DNA ligase (Takara, Japan) were used for construction of vectors. PCR reactions were performed using I-5™ 2 × High-Fidelity Master Mix DNA polymerase (TsingKe Biotech, Beijing, China). Plasmid isolations from E. coli cells were performed using the Plasmid Miniprep Kit (TsingKe Biotech, Beijing, China). Purification of DNA fragments from agarose gels or PCR reactions was carried out using Gel Extraction Kit (Omega, USA) and Cycle Pure Kit (Omega, USA), respectively. Primers were synthesized by TsingKe (China).

**Molecular cloning**. The DNA sequences that encode the two-subunit transketolase Rif15, and the separated subunits Rif15a and Rif15b were amplified from the genomic DNA of Amycolatopsis mediterranei U32 (A. mediterranei U32 was deposited in Institute of Microbiology, Chinese Academy of Sciences designated as CGMCC4.5720) under standard PCR conditions using the primer pairs of rif15-F/ rif15-R, rif15a-pSJ2F/rif15a-pSJ2R, and rif15b-F/rif15b-R, respectively (Supplementary Table 4). The P450 gene rif16 and the mutant gene rif16$_{R84W}$ were amplified from A. mediterranei S699 and U32 (Professor Guoping Zhao's laboratory collection) gDNA, respectively, using a pair of primers rif16-F and rif16-R (Supplementary Table 4). The rif15a fragment was double digested by BamHI/ HindIII and cloned into the expression vector pSJ2 (a derivative of pET21a, which is a gift from Professor Jiahai Zhou at Shanghai Institute of Organic Chemistry, Chinese Academy of Sciences) using standard molecular cloning methods to generate pSJ2-rif15a (Supplementary Fig. 2). The rif15, rif15b, rif16, and rif16$_{R84W}$ PCR products were double digested by NdeI/HindIII, NdeI/EcoRI, NdeI/XhoI, and NdeI/XhoI, respectively, and ligated into the corresponding pre-treated pET28b (Novagen, USA) to afford pET28b-rif15, pET28b-rif15b, pET28b-rif16, and pET28b-rif16$_{R84W}$ (Supplementary Fig. 2). All constructs were confirmed by DNA sequencing (Genewiz, China), and transformed into E. coli Codon Plus(DE3)-RIPL for expression of N-terminal His₆-tagged recombinant proteins.

**Protein expression and purification**. The E. coli Codon Plus(DE3)-RIPL transformant carrying pSJ2-rif15a was grown at 37 °C overnight in LB media supplemented with ampicillin (50 μg/mL), chloramphenicol (34 μg/mL), and

streptomycin (50 μg/mL). The transformant carrying pET28b-*rif15*, pET28b-*rif15b*, pET28b-*rif16* or pET28b-*rif16*$_{R84W}$ was cultured at 37 °C overnight in LB media supplemented with kanamycin (50 μg/mL), chloramphenicol (34 μg/mL), and streptomycin (50 μg/mL). The overnight seed culture was inoculated (1:100) into 1 L LB broth containing 10% glycerol, appropriate selective antibiotics and rare salt solution (6.75 mg/L FeCl₃, 500 μg/L ZnCl₂, CoCl₂, Na₂MoO₄, 250 μg/l CaCl₂, 465 μg/l CuSO₄, and 125 μg/l H₃BO₃), and then cultured at 37 °C until OD$_{600}$ reached 0.6–1 (~3–4 h). Next, 0.1 mM isopropyl β-D-thiogalactoside (IPTG, Sigma, USA) was added to induce protein expression, and thiamin (1 mM, Sigma, USA) and δ-aminolevulinic acid (1 mM, Sigma, USA) were supplemented to support efficient expression of *holo*-form P450 enzymes. The cells were grown at 16 °C for 20 h. All recombinant proteins including Rif15, Rif15a, Rif15b, Rif16, and Rif16$_{R84W}$ were purified by Ni-NTA affinity chromatography[42]. Briefly, *E. coli* cells were collected by centrifugation at 5000 ×*g* for 5 min, the cell pellet was re-suspended with 20 mL lysis buffer (20 mM Tris-HCl, 300 mM NaCl, 10% (w/v) glycerol, and 10 mM imidazole, pH 8.0), and the cells were broken by sonication (5 s on/5 s off) for 30 min on ice. Next, the cellular debris was removed by centrifugation at 12,000 ×*g* for 30 min. To the supernatant 1 mL nickel-nitrilotriacetic acid resin (Qiagen, Germany) was added, and each mixture was incubated for 30 min with gentle shaking at 4 °C. The resin was loaded onto an empty column and washed with wash buffer (20 mM Tris-HCl, 300 mM NaCl, 10% (w/v) glycerol, and 20 mM imidazole, pH 8.0) until no protein could be detected in the flow-through. The proteins bound to resin were eluted by 10 mL elution buffer (20 mM Tris-HCl, 300 mM NaCl, 10% (w/v) glycerol, and 250 mM imidazole, pH 8.0). Finally, the eluents were concentrated and buffer-exchanged to reaction buffer (20 mM Tris-HCl, 10% (w/v) glycerol, pH 7.4) via repeated ultrafiltration using Amicon Ultra-15 centrifugal filter units (Millipore, Ireland) with a 10-kDa cutoff. All protein purification steps were performed at 4 °C. The SDS-PAGE analysis (Supplementary Fig. 3) showed that Rif15, Rif15a, and Rif15b were >95% pure. However, the purity of Rif16 and Rif16$_{R84W}$ were unsatisfactory. Therefore, the collected Rif16 and Rif16$_{R84W}$ proteins were buffer-exchanged again to 20 mM Tris-HCl buffer (pH 8.0) by repeated ultra-filtration using Amicon Ultra-15 centrifugal filter units (Millipore, Ireland) with a 30-kDa cutoff. The buffer-exchanged proteins were further loaded onto a Mono Q 5/50 GL column (GE Healthcare, USA) pre-equilibrated with 20 mM Tris-HCl buffer (pH 8.0), and eluted with a gradient volume of 20 CV and an increasing ionic strength up to 1 M NaCl at a flow rate of 1 mL/min using ÄKTA purifier-100 (GE Healthcare, USA). Fractions containing proteins of interest were flash-frozen by liquid nitrogen and stored at −80 °C for later use.

**Protein concentration determination.** For Rif16 and Rif16$_{R84W}$, the UV-visible spectra were recorded on a DU 800 spectrophotometer (Beckman Coulter, USA). The CO-bound reduced difference spectrum was employed to determine the functional concentration of P450 enzymes using the extinction coefficient ($\varepsilon_{450nm-490nm}$) of 91,000 M$^{-1}$·cm$^{-1}$[43]. Briefly, CO was slowly bubbled through the Na₂S₂O₄ reduced P450 enzyme solution using a Pasteur pipette in a fume hood. The spectra of ferric, CO-bound, and CO-bound reduced forms of the P450 enzyme were recorded between 250 and 550 nm for generation of the CO-bound reduced difference spectrum. The protein concentrations of other proteins were determined using the Bradford assay with bovine serum albumin as standard[44].

**Spectral substrate binding assays.** Spectral substrate binding assays were carried out on a UV-visible spectrophotometer 50 Bio (Cary, USA) at room temperature by titrating 100 μM rifamycin L DMSO solution (blank DMSO for the reference group) into 1 mL of 1 μM Rif16 or Rif16$_{R84W}$ solution in 1 μL aliquots, leading to the substrate concentrations ranging from 0.1 to 1.2 μM. The series of Type I difference spectra were used to deduce $\Delta A$ ($A_{peak(390 nm)} - A_{trough(420 nm)}$). Then, the $\Delta A$ data versus substrate concentrations were fit to Michaelis–Menten equation to calculate the dissociation constant $K_d$[45].

**In vitro enzymatic assays of Rif15 and Rif16.** The Rif15 reaction mixture contained 10 μM Rif15a, 10 μM Rif15b, 200 μM R-S (or other rifamycin derivatives, National Institutes for Food and Drug Control, China), 0.5 mM ThDP (Sigma, USA), 2.5 mM MgCl₂, and 2 mM F-6-P (J&K Scientific, China) (or Xu-5-P, Ru-5-P, S-7-P, DHA from Sigma), in 100 μL of reaction buffer (20 mM Tris-HCl, 10% glycerol, pH 7.4). The Rif16 reaction assay was carried out in 100 μL of the same reaction buffer containing 2 μM Rif16, 200 μM R-L, 20 μM *se*Fdx plus 10 μM *se*FdR, and 1 mM NADPH (Solarbio, China) or 2 μM Rif16, 200 μM R-L, and 20 mM H₂O₂. The Rif15 and Rif16 reactions were incubated at 28 °C for 4 h and 1 h (unless otherwise specified), respectively, and quenched by mixing with the same volume of methanol. After high-speed centrifugation (20,000 ×*g*) for 15 min, the supernatants were analyzed on an Agilent 1260 infinity HPLC system (Agilent Technologies, USA) equipped with an ultraviolet detector. Compounds were separated by SB-C18 reverse-phase column (Thermo, 5 μm, 46 mm, USA) in a gradient system consisting of ddH₂O + 0.1% trifluoroacetic acid as solvent A and acetonitrile as solvent B. The program of solvent gradient is as follow: 40% B for 5 min, 40-80% B over 15 min, and 80–100% B over 5 min at a flow rate of 1 mL/min. The wavelength of detection was 425 nm for all rifamycin derivatives except for R-O (370 nm). The peak identity in each HPLC trace was assigned by comparison with authentic standards regarding the retention time and the

UV-visible spectrum, and confirmed by co-injection experiments and LC-HRMS analysis.

**Kinetic analysis of Rif15.** The kinetic analysis of Rif15 was carried out using 2–10 μM Rif15a/Rif15b, 10–100 μM R-S as substrate, 2 mM F-6-P as the C₂ keto donor, 2.5 mM MgCl₂, and 0.5 mM ThDP in 100 μL of reaction buffer. The reactions were performed 15–30 min at 28 °C and quenched by mixing with the same volume of methanol. After high-speed centrifugation (20,000 ×*g*) for 15 min, the supernatants were analyzed on an Agilent 1260 infinity HPLC system as described above. Each peak area of R-L was used to calculate the product concentration based on the standard curve of R-L. The triplicated data were fit to the Michaelis–Menten equation to determine the $k_{cat}$ and $K_m$ values using Origin 9.0.

**Preparation of ¹³C labeled F-6-P, R-L, and R-B.** To prepare [1-¹³C]-F-6-P, the reaction mixture containing 5 mM [1-¹³C] glucose (Sigma, USA), 6.25 mM MgCl₂, 12.5 mM ATP, 1 U hexokinase (Sigma, USA), 1 U phosphoglucose isomerase (Sigma, USA) was incubated at 28 °C for 16 h[46,47]. To prepare the ¹³C-labeled R-L, 5 mM [1-¹³C]-glucose, 8.75 mM MgCl₂, 12.5 mM ATP, 1 U hexokinase, 1 U phosphoglucose isomerase, 10 μM Rif15a, 10 μM Rif15b, 200 μM R-S, and 0.5 mM ThDP were mixed and incubated at 28 °C for 16 h. To prepare the ¹³C-labeled rifamycin B, [39-¹³C] R-L was first prepared as described above; after 8 h, 2 μM Rif16, 20 μM *se*Fdx, 10 μM *se*FdR, and 1 mM NADPH were added into the reaction mixture, and the reaction continued for 16 h at 28 °C. Notably, the one-pot reaction was unsuccessful because NADPH would reduce R-S to R-SV, thus preventing the Rif15 catalyzed reaction.

**Isolation and purification of R-L and R-B.** The scaled-up Rif15 (for R-L) and Rif16 (for R-B) reactions (20–100 mL) were respectively extracted by the same volume of ethyl acetate five times. Then, the extracts were dried under nitrogen flow, and re-dissolved in methanol. The purification of R-L or R-B was performed using semi-preparative HPLC (Waters XBridge™ Prep C18 5 μm, 10 × 250 mm) with a linear gradient of 40-80% acetonitrile in ddH₂O + 0.1% trifluoroacetic acid over 20 min, and then 100% acetonitrile for 5 min at a flow rate of 2.5 mL/min. The collected fractions containing R-L or R-B were combined. The solvents were removed using a Rotavapor R-3 rotary evaporator (Buchi, Switzerland) and N₂ blowing. Finally, 10.0 mg R-L and 3.0 mg R-B with > 98.0% purity were obtained.

**LC-HRMS analysis.** The LC-HRMS analysis was performed on a Waters symmetry column (4.6 × 150 mm, RP18) using the negative-mode electrospray ionization with a linear gradient of 10–100% acetonitrile in ddH₂O with 0.1% formic acid over 20 min, and followed by 100% acetonitrile for 5 min at a flow rate of 0.5 mL/min. The high resolution mass spectra were recorded on a Dionex Ultimate 3000 coupled to a Bruker Maxis Q-TOF.

**NMR analysis.** The ¹³C NMR spectra of ¹³C-labeled glucose, G-6-P, and F-6-P were acquired using D₂O as solvent. The ¹H, ¹³C and 2D NMR spectra of rifamycin derivatives were obtained using CDCl₃ or MeOD as solvent on a Bruker Avance III 600 MHz spectrometer with a 5 mm TCI cryoprobe.

**Crystallization and structure determination.** The crystal of Rif16 alone and the complex crystal of Rif16 and R-L were obtained at 16 °C by hanging drop vapor diffusion. The native Rif16 crystal screen droplets consisted of a 1:1 (v/v) protein at 16 mg/mL and the well solution of 100 mM Bis-Tris, pH 6.5, 200 mM magnesium chloride hexahydrate, and 25% PEG3350. Rif16 and R-L were mixed at a molar ratio of 1:5 and the co-crystallization was carried out in 200 mM magnesium chloride hexahydrate, 100 mM Tris, pH 8.5, 30% PEG4000. Crystals appeared after 1 week and were ready for data collection in 20 days. The crystals were flash-cooled in liquid nitrogen. The diffraction data were collected at 100 K under the synchrotron radiation at beamline BL19U1 of the Shanghai Synchrotron Radiation Facility (SSRF). The data sets were integrated and scaled with the HKL3000 package[48]. The structure is determined by molecular replacement with the structure of CYP105A (PDB accession code 4OQS) as the initial search model with the program Phaser. The programs Refmac5 and Coot9 were used for the refinement and model building[49–52]. Ramachandran plots were generated with Coot9. The statistics for data processing and structure refinement are shown in Supplementary Table 5. The coordinates were deposited to Protein Data Bank and the PDB ID codes are 5YSM and 5YSW for substrate-free Rif16 and R-L-bound Rif16, respectively. Figures were prepared using PYMOL (http://www.pymol.org).

**Data availability.** Data that support the findings of this study have been deposited in Protein Data Bank with the PDB ID codes 5YSM and 5YSW. Data referenced in this study are available in GenBank with the accession codes AMED_0651, AMED_0652, and AMED_0653. All other relevant data are available from the corresponding authors.

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

## Acknowledgements

This work was supported by the Shandong Provincial Natural Science Foundation (ZR2017ZB0207 to W.Z. and S.L.), the National Natural Science Foundation of China (grants 81741155, 31422002, and 21472204 to S.L., 31600036 to F.Q., 21572243 to Y.X., 31430004 and 31670058 to G.Z.), Chinese Academy of Sciences (grants QYZDB-SSW-SMC042 to S.L. and XDPB0402 to Y.X.), and the Science and Technology Commission of

Shanghai Municipality (grant 15JC1400402 to Y.X.). We thank the staff of BL19U1 beamline at the Shanghai Synchrotron Radiation Facility for assistance during X-ray diffraction data collection. We are also grateful to Dr. Gong Chen at Nankai University for helpful discussions about reaction mechanisms.

## Author contributions

F.Q., C.L., J.W., G.-L.T., Y.X., G.Z., and S.L. designed research; F.Q., C.L., F.L., X.Z., J.W., W.Z., Z.F., and W.L. performed research; F.Q., C.L., F.L., X.Z., J.W., G.-L.T., Y.X., G.Z., and S.L. analyzed results; F.Q., C.L., F.L., X.Z., Y.X., G.Z., and S.L. wrote the manuscript; All authors read and approved the manuscript.

## Additional information

**Competing interests:** The authors declare no competing interests.

