## [Peer Review File · Nature Communications]

REVIEWERS' COMMENTS:

Reviewer #1 (Remarks to the Author):

The authors have characterized two unique enzymes Rif15a/Rif15b and Rif16, which are involved in rifamycin biosynthesis. These enzymatic transformations are required to convert rifamycin S to rifamycin O via rifamycin L. Rif15a/Rif15b uses the thiamin chemistry to introduce the glycolate moiety into rifamycin S with a unique rearrangement reaction to afford rifamycin L. Rif16 is an atypical cytochrome P450 enzyme and catalyzes the transformation of ester to ether. Crystal structural analysis of Rif16 with rifamycin L as ligand showed a plausible recognition mechanism of substrate and then the authors could propose a plausible reaction mechanism of Rif16. Further, mutational analysis of Rif16 revealed that R84 is a critical amino acid residue to recognize rifamycin L as substrate. According to the previous reviewers' comments, they revised their manuscript properly. Thus, this manuscript would be now acceptable for publication.

Reviewer #2 (Remarks to the Author):

rifamycin synthesis.

The authors have added extra information requested by the referees which in general has improved the manuscript. The chemistry is good but some of the discussion around the structure and sequence comparisons falls short of the level I think is required for this journal.

Figure S16 The sequence of rif16 has been aligned with various disparate P450s. While this is useful it would be more informative if some of the P450s which were found in Table S3 were included. I also think they should include the closest structurally characterised P450 homolog (by sequence alignment) in this alignment (unless it is included already).

In the legend of Figure S16 shouldn't the physiological substrates of CYP51 and CYP199A4 be given these are not 4,4'-dihydroxybenzophenone and 1H-indole-6-carboxylic acid. I think they are lanosterol and 4-methoxybenzoic acid. 1,3,3-trimethyl-2-oxabicyclo[2.2.2]octane could be called 1,8-cineole (P450cin). Perhaps providing the references used would help resolve this matter.

Line 159 to 161 From the overlay of the two structures in Fig. 4 there are very few changes in the structure other than the now observable F/G loop. Is this what the authors mean by going from an open to partially open? Is there any evidence of other movements in the structure that are observable between the open and closed forms of other P450s such as CYP101A1? Are there crystal contacts/artefacts in the structure holding the enzyme in the open form, as have reported in other structures. One important observation from these structures of open forms of P450 enzymes is that the substrate binds differently to that of the closed form. All of these aspects are impact on the study in enabling an assessment of the validity of the data used from the substrate bound structure presented here.

Table S3 the table was cropped so some data was missing (portrait not landscape) but it seemed to contain useful information. It would be useful if the authors could add if the Rif15 and Rif16 equivalents are found in each bacteria given and if they are collocated in the genome (As they appear to be based on the sequence ID of the species in *Amycolatopsis vancoresmycina*).

Minor comments modify lines 150 "one typical cytochrome P450 fold existing in an asymmetric unit."

Line 151 Remove the word remarkably

Line 153-154 Is this electron density missing in both the substrate-free and substrate bound forms?

Line 156 Change "Lacking a bound" to "in the absence of"

Line 163 replace 5YSW with substrate-bound

Line 176 change "reported previously" to "proposed"

Figure S20 What is the concentration of the P450 used in the assay.

Line 208 some introduction to the fact the authors have cloned and purified this mutant would be useful. This information is currently hidden in the experimental section.

Line 318-319 Four spectra are listed as being recorded but only three are shown in the figures in the supporting info. Is this correct?

Reviewer #3 (Remarks to the Author):

The authors did everything what could be reasonably expected to fulfill the reviewers requests. This should be now accepted.

Responses to Reviewers (Authors' responses to reviewers are in red)

Reviewers Comments:

Referee #1 (Remarks to the Author):

The authors have characterized two unique enzymes Rif15a/Rif15b and Rif16, which are involved in rifamycin biosynthesis. These enzymatic transformations are required to convert rifamycin S to rifamycin O via rifamycin L. Rif15a/Rif15b uses the thiamin chemistry to introduce the glycolate moiety into rifamycin S with a unique rearrangement reaction to afford rifamycin L. Rif16 is an atypical cytochrome P450 enzyme and catalyzes the transformation of ester to ether. Crystal structural analysis of Rif16 with rifamycin L as ligand showed a plausible recognition mechanism of substrate and then the authors could propose a plausible reaction mechanism of Rif16. Further, mutational analysis of Rif16 revealed that R84 is a critical amino acid residue to recognize rifamycin L as substrate. According to the previous reviewers' comments, they revised their manuscript properly. Thus, this manuscript would be now acceptable for publication.

We appreciate the recommendation of acceptance from this reviewer.

Reviewer #2 (Remarks to the Author):

rifamycin synthesis.

The authors have added extra information requested by the referees which in general has improved the manuscript. The chemistry is good but some of the discussion around the structure and sequence comparisons falls short of the level I think is required for this journal.

1. Figure S16 The sequence of rif16 has been aligned with various disparate P450s. While this is useful it would be more informative if some of the P450s which were found in Table S3 were included. I also think they should include the closest structurally characterised P450 homolog (by sequence alignment) in this alignment (unless it is included already).

The purpose of Supplementary Fig. 16 is to support the statement of "The BB' loop-B' helix-B'C loop region, which is known to be important for substrate specificity determination, is significantly longer than those of many P450 enzymes that recognize smaller substrates" (lines 152-154). However, to address this reasonable suggestion, Supplementary Fig. 24 with the protein sequence alignment of Rif16 and a select number of its analogous P450 enzymes has been added in the Supplementary Information.

2. In the legend of Figure S16 shouldn't the physiological substrates of CYP51 and CYP199A4 be given these are not 4,4'-dihydroxybenzophenone and 1H-indole-6-carboxylic acid. I think they are lanosterol and 4-methoxybenzoic acid. 1,3,3-trimethyl-2-oxabicyclo[2.2.2]octane could be called 1,8-cineole (P450cin). Perhaps providing the references used would help resolve this matter.

In Supplementary Fig. 16, CYP51 is actually from *Mycobacterium tuberculosis*, rather than the eukaryotic cytochrome P450 14 α -sterol demethylase (also called CYP51) that uses lanosterol as substrate. Although the physiological substrate of CYP51_{Mt} remains unknown, its crystal structures in complex with some small ligands such as 4,4'-dihydroxybenzophenone (m.w. 214.2) and estriol (m.w. 288.38) are available in PDB. 4-Methoxybenzoic acid (m.w. 152.15) is indeed the physiological substrate of CYP199A4. Therefore, we have corrected this in the revised figure legend. As pointed out by this reviewer, 1,3,3-trimethyl-2-oxabicyclo[2.2.2]octane can also be called 1,8-cineole, and we used this more concise name as suggested. To avoid any unnecessary confusions, we have provided the references as suggested.

3. Line 159 to 161 From the overlay of the two structures in Fig. 4 there are very few changes in the structure other than the now observable F/G loop. Is this what the authors mean by going from an open to partially open? Is there any evidence of other movements in the structure that are observable between the open and closed forms of other P450s such as CYP101A1? Are there crystal contacts/artefacts in the structure holding the enzyme in the open form, as have reported in other structures. One important observation from these structures of open forms of P450 enzymes is that the substrate binds differently to that of the closed form. All of these aspects are impact on the study in enabling an assessment of the validity of the data used from the substrate bound structure presented here.

The open form means that in the substrate free structure, the B' and F/G loop regions are disordered, thus missing the electron density (Fig. 4a). Upon substrate binding, the F/G loop becomes ordered, but the B' region remains disordered, thereby defining a partially open form (Fig. 4b). Besides the significant conformational changes in the F/G loop region (Fig. 4c) and probably also in the B' region that lacks electron density, there are only some slight movements in the substrate free and substrate bound structures. In addition, we do not see any evidence that crystal contacts/artefacts may hold the enzyme in the open form. Finally, we are confident of the validity of the structural data since they are well consistent with our biochemical results and the mechanisms supported by the ¹³C-tracer NMR experiments.

4. Table S3 the table was cropped so some data was missing (portrait not landscape) but it seemed to contain useful information. It would be useful if the authors could add if the Rif15 and Rif16 equivalents are found in each bacteria given and if they are collocated in the genome (As they appear to be based on the sequence ID of the species in *Amycolatopsis vancoresmycina*).

Supplementary Table 3 has been revised as suggested.

Minor comments modify
lines 150 “one typical cytochrome P450 fold existing in an asymmetric unit.”

We do not think this sentence is problematic.

Line 151 Remove the word remarkably

Removed (line 152).

Line 153-154 Is this electron density missing in both the substrate-free and substrate bound forms?

Yes. This has been clarified in the revised manuscript (line154-155).

Line 156 Change “Lacking a bound” to “in the absence of”

Corrected as suggested (line 157-158).

Line 163 replace 5YSW with substrate-bound

Replaced as suggested (line 164).

Line 176 change “reported previously” to “proposed”

Changed as suggested (line 177).

Figure S20 What is the concentration of the P450 used in the assay.

The concentration of Rif16 (1 μ M) has been added as suggested.

Line 208 some introduction to the fact the authors have cloned and purified this mutant would be useful. This information is currently hidden in the experimental section.

Since the Rif16_{R84W} mutant was purified using the same procedure as wild-type enzyme, we choose not to repeat this information that is introduced earlier in lines 76-79 and described later in lines 269-304.

Line 318-319 Four spectra are listed as being recorded but only three are shown in the figures in the supporting info. Is this correct?

Revised as suggested (line 310).

Reviewer #3 (Remarks to the Author):

The authors did everything what could be reasonably expected to fulfill the reviewers requests. This should be now accepted.

We appreciate the recommendation of acceptance from this reviewer.